# A New Repellent for Redbay Ambrosia Beetle (Coleoptera: Curculionidae: Scolytinae), Primary Vector of the Mycopathogen That Causes Laurel Wilt

**DOI:** 10.3390/plants12132406

**Published:** 2023-06-21

**Authors:** Kevin R. Cloonan, Wayne S. Montgomery, Teresa I. Narvaez, Paul E. Kendra

**Affiliations:** USDA-ARS, Subtropical Horticulture Research Station, Miami, FL 33158, USA; wayne.montgomery@usda.gov (W.S.M.); isabelt87@hotmail.com (T.I.N.); paul.kendra@usda.gov (P.E.K.)

**Keywords:** *Harringtonia lauricola*, invasive species, Lauraceae, piperitone, *Xyleborus glabratus*

## Abstract

The redbay ambrosia beetle, *Xyleborus glabratus*, was detected in Georgia, USA, in 2002 and has since spread to 11 additional states. This wood-boring weevil carries a symbiotic fungus, *Harringtonia lauricola*, that causes laurel wilt, a lethal disease of trees in the Lauraceae family. Native ambrosia beetles that breed in infected trees can acquire *H. lauricola* and contribute to the spread of laurel wilt. Since 2002, laurel wilt has devastated native *Persea* species in coastal forests and has killed an estimated 200,000 avocado trees in Florida. Since laurel wilt is difficult to manage once it has entered a susceptible agrosystem, this study evaluated piperitone as a candidate repellent to deter attacks by *X. glabratus* and other ambrosia beetles. Additionally, piperitone was compared to the known repellent verbenone as a potential cost-effective alternative. The repellent efficacy was determined by comparing captures in traps baited with commercial beetle lures containing α-copaene versus captures in traps baited with lures plus a repellent. In parallel 10-week field tests, the addition of piperitone reduced the captures of *X. glabratus* in α-copaene-baited traps by 90%; however, there was no significant reduction in the captures of native ambrosia beetles in ethanol-baited traps. In two replicate 10-week comparative tests, piperitone and verbenone both reduced *X. glabratus* captures by 68–90%, with longevity over the full 10 weeks. This study identifies piperitone as a new *X. glabratus* repellent with potential for pest management.

## 1. Introduction

Exotic ambrosia beetles are commonly intercepted at ports-of-entry in the USA [1,2,3,4], though they are rarely elevated to agricultural or forestry pests. In 2002, however, the invasive redbay ambrosia beetle, *Xyleborus glabratus* Eichhoff (Coleoptera: Curculionidae: Scolytinae: Xyleborini) (Figure 1A,B), was detected for the first time in North America in Port Wentworth, Georgia [2,5]. *Xyleborus glabratus* is native to Southeast Asia [6] and, unlike other ambrosia beetle species that colonize dying or dead trees [7] and are considered beneficial to forest ecosystems [8], *X. glabratus* can attack healthy, unstressed trees [9].

Most ambrosia beetles feed exclusively on symbiotic fungi that adult beetles purposefully cultivate inside their brood galleries [10]. Female *X. glabratus,* like other ambrosia beetles [11,12,13,14,15], are the only winged sex and carry the conidia of these fungal symbionts in cuticular pouches at the base of their mandibles, called mycangia [9]. During gallery excavation (Figure 1C), females inoculate the host xylem with these symbiotic fungi that then provide nutrition for developing larvae and adults; the host wood is not consumed [16]. One of these fungal symbionts, *Harringtonia lauricola* T.C. Harr., Fraedrich & Aghayeva (Ophiostamatales: Ophiostomataceae) (previously *Raffaelea lauricola* [17]), is the causal agent of laurel wilt, a systemic vascular disease of trees and shrubs in the family Lauraceae (Figure 1D) [9,18].

*Harringtonia lauricola* infection does not directly kill the host tree. Infected trees launch a series of defensive responses, including the secretion of resins and formation of tyloses (parenchymal cell outgrowth into the xylem vessels), in an attempt to restrict the movement of *H. lauricola* [19]. This defensive response blocks the xylem vessels and impedes water transport [20], causing tree death in as little as 4 to 6 weeks after initial infection [21]. Trees in the genus *Persea* are particularly susceptible to laurel wilt disease (see Kendra et al. [22] for a list of known suscepts in the USA), including those important in forest ecosystems such as redbay, *Persea borbonia* (L.) Spreng [9,23], swamp bay, *P. palustris* (Raf.) Sarg. [24,25], and silkbay, *P. humilis* Nash [26], as well as the economically important agricultural commodity avocado, *P. americana* Mill [27].

Laurel wilt has spread rapidly throughout the Atlantic and Gulf Coastal Plains [28,29,30,31,32,33,34,35] and, as of October 2022, had been detected in 12 US states [36]. To date, laurel wilt is estimated to have killed 300,000 redbay trees [37] and 200,000 avocado trees in Miami-Dade county, Florida [21,38,39,40,41,42], and is responsible for more than 90 percent of the mortality among susceptible trees in some forested areas of the USA [9,43,44,45,46]. The rapid and widespread movement of laurel wilt does not correlate with *X. glabratus* captures or detection in agrosystems affected by laurel wilt [47,48,49,50], suggesting that native ambrosia beetle species have acquired the fungal symbiont via lateral transfer from *X. glabratus* or host trees infected with *H. lauricola* [51,52,53,54,55].

Since the initial introduction of *X. glabratus* into the USA in 2002, *H. lauricola* has been recovered from nine additional ambrosia beetle species in Florida, including *Ambrosiodmus lecontei* Hopkins, *Xylosandrus crassiusculus* (Motschulsky), *Xyleborinus andrewesi* (Blandford), *Xyleborinus gracilis* (Eichhoff), *Xyleborinus saxesenii* (Ratzeburg), *Xyleborus affinis* Eichhoff, *Xyleborus bispinatus* Eichhoff, *Xyleborus volvulus* (Fabricius), and *Xyleborus ferrugineus* (Fabricius) [51,53,54,55,56]. Three species, *X. bispinatus* [56], *X. volvulus*, and *X. ferrugineus* [51], have been experimentally shown to transmit *H. lauricola* spores to healthy avocado trees and induce laurel wilt. Moreover, laboratory-reared *X. bispinatus* have been shown to survive and reproduce on a sole diet of *H. lauricola*, and this mutualism can persist throughout several generations [57]. Thus, monitoring and controlling *X. glabratus* as well as the secondary vectors of *H. lauricola* are important aspects of IPM programs for managing laurel wilt.

Lure development for *X. glabratus* is based on early studies showing that *X. glabratus*, as a primary colonizer attacking healthy unstressed trees, is attracted to the volatile emissions of monoterpenes and sesquiterpenes from the host wood (see Martini et al. [58] and the references therein for a review of *X. glabratus* attractants), specifically, α-copaene, β-caryophyllene, eucalyptol, α-humulene, and δ-cadinene [27,59,60,61,62,63]. Currently, the most attractive synthetic lure for *X. glabratus* consists of a distilled essential oil product containing 50% (-)-α-copaene [64,65]. This 50% (-)-α-copaene lure is used in monitoring and surveillance programs for *X. glabratus*, and ethanol lures are the standard used for monitoring overall ambrosia beetle populations [66]. Although effective monitoring lures are available for *X. glabratus* and the secondary vectors of *H. lauricola*, few management tools are available for controlling laurel wilt in natural or agricultural ecosystems. Due to the difficulty in curtailing laurel wilt once it is established in a susceptible environment [67] and the likely incursion of *X. glabratus* into Mexico, which contains suscepts of laurel wilt [22,68], more tools for managing *X. glabratus* and the secondary vectors of laurel wilt are needed.

Repellents have been used in IPM programs for controlling some bark beetles in forestry settings since the 1980s. For example, verbenone, a known bark beetle repellent, was originally utilized to manage *Dendroctonus* bark beetle species [69,70,71,72,73] and is used to protect lodgepole pine, *Pinus contorta* var. *latifolia* Engelm, against infestation from the mountain pine beetle *Dendroctonus ponderosae* Hopkins (Coleoptera: Curculionidae: Scolytinae: Hylesinini) [74,75]. Verbenone also reduced Xyleborine beetle captures in lure-baited traps for *Xylosandrus compactus* (Eichhoff), *X. crassiusculus* (Motschulsky), *X. germanus* (Blandford), *Xyleborinus saxesenii* (Ratzeburg) [76,77,78], and *X. glabratus* [79,80]. Although an effective beetle repellent, verbenone is expensive, and alternative candidate repellents should be explored.

Piperitone (*p*-menth-1-en-3-one), a monoterpene ketone that is similar in structure to verbenone, has been shown to repel Xyleborine beetles in the *Euwallacea* nr. *fornicatus* species complex, including the Kuroshio shot hole borer, *E. kuroshio* Gomez and Hulcr [81]; the polyphagous shot hole borer, *E. fornicatus* Eichhoff [82,83]; and the tea shot hole borer, *E. perbrevis* Schedl [84]. As a potential cost-effective alternative to verbenone, piperitone was evaluated as a candidate repellent for *X. glabratus* and other ambrosia beetle species through a series of six 10-week field tests conducted in South Florida. In each test, the efficacy of repellency was determined by comparing beetle captures in lure-baited traps versus captures in traps baited with lures plus a repellent. Commercial α-copaene lures were used in tests targeting *X. glabratus*; low-release ethanol lures were used in tests targeting the overall Scolytine community.

## 2. Results

### 2.1. Field Evaluations—Miami-Dade County

In field test 1 (Figure 2A), there were significant differences in the mean captures of *X. glabratus* among treatments (*F* = 7.068; df = 2,12; *p* = 0.017). Traps baited with the α-copaene lure plus a piperitone dispenser captured significantly fewer beetles than traps baited with the lure alone, and these captures were not significantly different from those intercepted with the non-baited control trap. The presence of piperitone resulted in a mean reduction in captures of 90.75 ± 6.19%.

In field test 2 (Figure 2B), there were significant differences in the mean captures of bark and ambrosia beetles (all species combined, excluding *X. glabratus*) among treatments (*F* = 32.393; df = 2,12; *p* < 0.001). Traps baited with the ethanol lure alone or ethanol + piperitone captured significantly more beetles than the non-baited controls. The addition of piperitone reduced captures by 28.32 ± 9.09%, but this difference was not statistically significant.

Combining the sampling efforts from both field tests, we detected at least 18 species within the subfamily Scolytinae (*Hypothenemus* were grouped as a genus, not identified to the species level) in the tree island ecosystem, with the majority representative of the tribe Xyleborini (Table 1). This included all ten species from which the laurel wilt pathogen, *H. lauricola*, has been isolated [55]. In field test 1 with α-copaene lures, a total of 312 specimens were collected. *Xyleborus glabratus* comprised the highest number of captures (43.3%), followed by *Xyleborus volvulus* (Fabricius) (11.2%), *Xyleborinus saxesenii* (Ratzeburg) (10.6%), and *Xylosandrus compactus* (Eichhoff) (7.1%). In field test 2 with ethanol lures, a total of 1625 specimens were collected, dominated by captures of *X. saxesenii* (27.6%), *Hypothenemus* spp. (23.4%), *Corthylus papulans* Eichhoff (14.5%), *X. volvulus* (13.1%), and *Xyleborus affinis* Eichhoff (6.9%).

### 2.2. Field Evaluations—Broward County

In field test 3 (Figure 3A), there were significant differences in the mean captures of *X. glabratus* among treatments (*F* = 10.350; df = 2,12; *p* = 0.002). Traps baited with the α-copaene lure plus piperitone captured significantly fewer beetles than traps baited with the lure alone, and these captures were not significantly different from those of the non-baited control trap. The presence of piperitone resulted in a mean reduction in captures of 68.28 ± 9.46%.

Likewise, in field test 4 (Figure 3B), there were significant differences in the mean captures of *X. glabratus* among treatments (*F* = 20.453; df = 2,12; *p* < 0.001). Consistent with field tests 1 and 3, traps baited with the α-copaene lure plus piperitone captured significantly fewer beetles than traps baited with the lure alone, and these captures were comparable to those of the non-baited control trap. The addition of piperitone achieved a reduction in captures of 73.15 ± 2.89%.

In field test 5 (Figure 4A), there were significant differences in the mean captures of *X. glabratus* among the four treatments (*F* = 44.176; df = 3,15; *p* < 0.001). The highest numbers were obtained in traps baited with the α-copaene lure alone. The captures with the lure plus piperitone or the lure plus verbenone were significantly lower than those with the lure alone but not different from those obtained with the non-baited control trap. The presence of piperitone resulted in a mean reduction in captures of 68.61 ± 2.84%, and the presence of verbenone resulted in an equivalent reduction of 68.96 ± 5.15%. Examination of the weekly captures (Figure 4B) indicated that the decrease was significant up through week 10 for both piperitone (*t* = 4.525, df = 8, *p* = 0.002) and verbenone (*t* = 4.428, df = 8, *p* = 0.002).

In field test 6 (Figure 4C), there were significant differences in the mean captures of *X. glabratus* among the four treatments (*F* = 33.322; df = 3,15; *p* < 0.001). As observed in test 5, traps baited with the α-copaene lure alone caught the highest number of beetles. The traps baited with the lure plus either piperitone or verbenone caught significantly fewer beetles than the lure alone, and these captures were not significantly different from the non-baited control. The addition of piperitone resulted in a mean decrease in captures of 79.54 ± 6.12%, and the addition of verbenone resulted in a decrease of 73.96 ± 4.86%. The weekly capture data (Figure 4D) showed that the repellent effect was significant up through week 10 for both piperitone (*t* = 2.761, df = 8, *p* = 0.024) and verbenone (*t* = 3.029, df = 8, *p* = 0.016).

## 3. Discussion

Laurel wilt disease is a serious threat to commercial avocado groves in Florida [38,40,41,85] and forested areas containing susceptible lauraceous trees. Members of the Lauraceae family are important to forest ecosystems, as they provide food for wildlife during the autumn and winter months [86] and are the sole food source for some herbivores, such as the Palamedes swallowtail butterfly *Papilio palamedes* Drury (Lepidoptera: Papilionidae) [87]. Increased light penetration due to overstory loss from dead trees can also alter forest productivity [46]. The rate of spread of laurel wilt in ecosystems containing redbay and sassafras trees in the southeastern United States slowed from 40 km/year in 2016 to 24 km/year in 2021, likely due to the exhaustion of susceptible hosts [88]. To prevent the complete loss of susceptible Lauraceae in affected areas and prevent further spread to unaffected areas such as California [89] and Mexico [22], more control tools are needed as part of an integrated pest management (IPM) approach for controlling *X. glabratus* and other ambrosia beetle vectors of *H. lauricola*. Repellent compounds that prevent initial beetle infestation may offer some protection to agrosystems impacted by laurel wilt.

The current study tested piperitone as a candidate repellent for *X. glabratus* and other ambrosia beetles in swampbay forests with laurel wilt. In field tests 1 (Figure 2A), 3 (Figure 3A), and 4 (Figure 3B), the addition of piperitone to α-copaene-baited traps reduced *X. glabratus* captures by 90%, 68%, and 73%, respectively. Previous studies identified other compounds as repellents of *X. glabratus*, including myrcene, camphene, methyl salicylate, and verbenone [79]. Hughes et al. [79] showed that verbenone, applied to freshly cut redbay bolts via a slow-release, inert, wax-like matrix called ‘Specialized Pheromone and Lure Application Technology’ (SPLAT) (ISCA Technologies, Riverside, CA, USA), reduced *X. glabratus* landings and borings compared to untreated bolts. SPLAT–verbenone applied directly to live standing redbay trees also reduced *X. glabratus* landing and boring incidence [80] and increased tree survival rates from 41.2% to 70.2% [80]. These results are promising, and more work should explore factors affecting the efficacy of verbenone as an IPM tool for managing ambrosia beetle species that are a vector for *H. lauricola*. Verbenone is expensive to produce, however, and cheaper beetle repellents should be explored. Thus, the current study directly compared piperitone to verbenone for its efficacy and longevity as an *X. glabratus* repellent.

In field trial 5, piperitone and verbenone both reduced *X. glabratus* captures by 68%; in field trial 6, piperitone and verbenone reduced trap captures by 79 and 73%, respectively; and in both trials, the repellent effects lasted for the full 10-weeks. In studies using comparable field test designs, piperitone and verbenone have also been shown to repel ambrosia beetles in the *Euwallacea* nr. *fornicatus* species complex [81,82,83,84]. Like *X. glabratus*, these *Euwallacea* species are primary colonizers that infest live host trees [49]. An additional goal of the current study was to investigate the effects of piperitone on other, non-*X. glabratus* ambrosia beetles that utilize ethanol as a host location cue to colonize stressed and/or dying trees. In field trial 2 (Figure 2B), the addition of piperitone to ethanol-baited traps did reduce the overall bark and ambrosia beetle captures (excluding *X. glabratus*) by 28%, though this reduction was not significant.

Similar to previous trapping studies [52], most of the Xyleborini beetles captured in the ethanol-baited traps were *X. saxesenii*, *X. volvulus*, *X. affinis*, *X. andrewesi*, and *X. bispinatus* (Table 1), all of which are potential *H. lauricola* vectors [55]. Most ambrosia beetles, including those not repelled by piperitone in this study (Figure 2B) and those not repelled by verbenone in previous studies [90], colonize stressed or dying trees [7]. Ethanol is emitted as a byproduct of tree decay [66] and serves as an attractant for ambrosia beetle species that colonize dead or dying wood, as it signals suitable host material [91]. For example, container-grown oak trees, *Quercus robur* L., injected with increasing concentrations of ethanol resulted in greater attraction and boring rates for *X. saxesenii*, *X. germanus*, and *X. crassiusculus* [92]. Alternatively, *X. glabratus* is repelled by ethanol [52,59,93] because *X. glabratus* functions as a primary colonizer that infests healthy trees [49,61].

Dispersing *X. glabratus* females are initially attracted to terpenoid emissions from healthy Lauraceae host wood, including α-copaene [23,52,59,63,94]. These pioneer females bore into host trees and inoculate them with *H. lauricola* spores, but these initial colonization attempts often fail to develop natal galleries [9]. After infection with *H. lauricola*, the leaves of infected host trees emit large quantities of methyl salicylate [95] which may initially repel further *X. glabratus* colonization [79]. As the disease progresses, however, methyl salicylate emissions decline, and the production of α-copaene in the leaf material, which is not typically emitted from healthy leaves, increases [95]. Decreasing methyl salicylate emissions and increasing α-copaene emissions then initiate mass *X. glabratus* attack [58]. When an infected and decaying tree begins to produce ethanol, those species that function as decomposers, such as *X. saxesenii*, *X. volvulus*, *X. affinis*, *X. andrewesi*, and *X. bispinatus*, become attracted to, and infest, the degraded, suitable host material. In addition to ethanol, stressed and dying trees may also emit piperitone and verbenone as byproducts of plant degradation [96], therefore repelling primary colonizers such as *X. glabratus* and members of the *E.* nr. *fornicatus* complex, while not repelling species that colonize declining trees, such as *X. saxesenii*, *X. volvulus*, *X. affinis*, *X. andrewesi*, and *X. bispinatus*.

Lindgren et al. [97] explored this hypothesis by studying bark beetle species with different host preferences and their responses to verbenone. *Dendroctonus ponderosae* (LeConte) requires healthy and live hosts, *Ipis latidens* (LeConte) and *Ipis pini* (Say) prefer recently dead hosts [98], and *Hylurgops porosus* (LeConte) and *Hylastes longicollis* (Swaine) prefer dead hosts [7]. Since verbenone is associated with degraded plant material [96], Lindgren et al. [97] postulated that the response to verbenone would vary according to species, depending upon their preferences for hosts in different physiological stages. The results from Lindgren et al. [97] supported this hypothesis, in that verbenone repelled *D. ponderosae*, captures of *Ipis latidens* and *Ipis pini* were gradually reduced with increasing verbenone doses, and neither *H. porosus* nor *H. longicollis* were repelled by verbenone. Piperitone and verbenone likely signal information about unsuitable hosts to primary colonizers of healthy trees such as *X. glabratus*, *E. perbrevis*, *E. fornicatus*, and *E. kuroshio* but do not indicate unsuitable host quality to decomposers such as *X. saxesenii*, *X. volvulus*, *X. affinis*, *X. andrewesi*, and *X. bispinatus.* Efforts should be made to identify candidate repellents for *X. saxesenii*, *X. volvulus*, *X. affinis*, *X. andrewesi*, and *X. bispinatus*, since managing laurel wilt disease will require the suppression of populations of *X. glabratus* and these secondary vectors.

Additional research efforts should investigate factors impacting the efficacy of piperitone as an *X. glabratus* repellent, including the release rate, placement location, and deployment rate for developing an optimal piperitone dispenser. For example, Byers et al. [83] found that the placement and release rates [82] of piperitone dispensers impacted the repellency of *E. fornicatus* in Israel. As a potential alternative to verbenone, SPLAT–piperitone applications should be explored as an *X. glabratus* management tool. Incorporating piperitone dispensers into a push–pull system as part of an IPM program is another potential approach for managing *X. glabratus*. Push–pull systems consist of a stimulus that repels a pest insect away from a commodity, coupled with a stimulus that attracts and lures the pest away from the commodity, where it can be removed [99]. Since commercially produced α-copaene lures are available for monitoring *X. glabratus*, future research should explore the use of these lures in tandem with piperitone dispensers to develop a push–pull strategy for managing *X. glabratus* populations.

## 4. Materials and Methods

### 4.1. Lures, Repellents, and Traps

All field tests for *X. glabratus* utilized a commercially available lure containing an essential oil enriched in α-copaene [64]. The lure consists of a 2.9 cm diameter plastic bubble loaded with 2.0 mL oil (product #3302; Synergy Semiochemicals Corp., Delta, BC, Canada). Our previous analyses indicated that this lure has a field longevity of 12 weeks, and the oil is comprised of >50% α-copaene, of which 99.9% is the negative enantiomer [100]. In a single test designed to sample the community of bark and ambrosia beetles, traps were baited with a low-release ethanol lure that contained 15 mL of the product in a 40 cm long, white, plastic sleeve (Contech Enterprises Inc., Victoria, BC, Canada). For research purposes, Synergy Semiochemicals formulated plastic bubble dispensers (2.9 cm diameter) containing 2.0 mL of piperitone. This potential repellent was compared to Synergy’s commercially formulated verbenone dispensers, which contain 0.98 g in a 2.9 cm diameter bubble (product #3414).

The traps were constructed from two white sticky panels (23 × 28 cm, Scentry wing trap bottoms, Great Lakes IPM, Vestaburg, MI, USA) suspended back-to-back from the end of a wire hook, as previously described [84]. Ethanol sleeves were attached to the wire stem above the panels and then secured with binder clips along the outer edge of the panel to prevent contact with the adhesive surface. Bubble lures and repellent dispensers were clipped to the wire just above the sticky cards. The final assembly was topped with an inverted clear plastic plate (24 cm diameter) to shield the dispensers from rain. Previous evaluations have shown this sticky trap design to be more effective than conventional Lindgren funnel traps for field trapping of *X. glabratus* [101] and other ambrosia beetles [48].

### 4.2. Field Evaluations—Miami-Dade County

Two 10-week field trapping experiments were conducted on a series of tree islands within the South Dade Wetlands Preserve [102], just east of Everglades National Park in southeastern Miami-Dade County, Florida. These isolated upland habitats, surrounded by sawgrass wet prairie communities, were dominated by woody shrubs and trees, including swampbay (*P. palustris*) exhibiting mid- to late-stage laurel wilt [103]. Five sites with clusters of affected swampbay trees (Figure 1D) were selected as trapping locations (GPS coordinates: 25°24.268 N, 80°27.732 W; 25°24.257 N, 80°27.817 W; 25°24.261 N, 80°27.830 W; 25°24.267 N, 80°29.849 W; and 25°24.259 N, 80°27.880 W).

Field test 1 was conducted from 22 October to 31 December 2019 to assess the efficacy of piperitone as a repellent for *X. glabratus*. The treatments included α-copaene, α-copaene plus piperitone, and a non-baited control trap to determine passive captures of inflight females. Field test 2 was conducted from 7 January to 19 March 2020 to assess the efficacy of piperitone as a general repellent for bark and ambrosia beetles. The treatments included ethanol, ethanol plus piperitone, and a non-baited control trap.

Both tests followed a randomized complete block design, with five replicate blocks at the tree island sites identified above. Each block consisted of a set of traps encircling a cluster of symptomatic swampbay trees. The traps were hung from non-host trees in well-shaded locations, ~1.5 m above ground with a minimum of 10 m spacing between adjacent traps. The tests were serviced every 2 weeks, and on each sampling date, the sticky panels were collected, new panels were deployed, and the trap positions were rotated sequentially within each block to minimize the positional effects on beetle captures.

### 4.3. Field Evaluations—Broward County

Four subsequent 10-week field trials were conducted at a site in Broward County, Florida, that had a higher population level of *X. glabratus*. It was a small conservation easement adjacent to a roadside canal west of the Fort Lauderdale Metropolitan area. The low-lying site was densely vegetated with subtropical native and invasive plant species, including royal palm (*Roystonea regia*), *Ficus* spp., *Acacia* spp., and a high number of mature swampbay trees exhibiting early stages of laurel wilt [103] (site coordinates 26°06.034 N, 80°21.744 W).

Field test 3 was conducted from 12 March to 21 May 2020, and field test 4 was conducted from 21 May to 30 July 2020. Both tests were run to obtain further data supporting the efficacy of piperitone as a repellent for *X. glabratus*. The treatments included α-copaene, α-copaene plus piperitone, and a non-baited control trap. Field tests 5 and 6 were conducted from 20 August to 29 October 2020 and 14 January to 25 March 2021, respectively. These latter tests were run to compare the efficacy of piperitone to that of verbenone and included the following treatments: α-copaene, α-copaene plus piperitone, α-copaene plus verbenone, and a non-baited control trap. Field tests 3–6 followed a randomized complete block design similar to that of tests 1 and 2; however, the replicate blocks consisted of a row of traps, and the blocks were aligned to form a rectangular grid. The traps were checked every 2 weeks, with the treatment positions rotated within every block during each servicing.

### 4.4. Sample Processing

The sticky panels collected from the field were taken to the USDA-ARS laboratory (Miami, FL, USA) for processing. All bark and ambrosia beetles were removed from the panels, soaked briefly in a histological clearing agent (Histo-clear II; National Diagnostics, Atlanta, GA, USA) to remove the adhesive, and stored in 70% ethanol. With the exception of *Hypothenemus* spp., all specimens were examined under a dissecting microscope and identified to the species level using taxonomic references [5,104,105].

### 4.5. Statistical Analysis

One-way analysis of variance (ANOVA) was used to test the effect of treatment on the mean captures (beetles/trap/week) in each field test. Significant ANOVAs were then followed by mean separation with the Tukey HSD test. When necessary, the capture data were square-root (*x* + 0.05)-transformed to stabilize variance prior to ANOVA. Analysis using the *t*-test was used to compare captures with and without repellent on each sampling date in order to determine the longevity of repellency [84]. Analyses were performed using SigmaPlot 14.0 (Systat Software Inc., San Jose, CA, USA). Results are presented as the mean ± SEM; probability was considered significant at a critical level of α = 0.05.

## 5. Conclusions

The invasive redbay ambrosia beetle, *X. glabratus*, is the primary vector of *H. lauricola*, the mycopathogen that causes laurel wilt. To date, at least nine other ambrosia beetle species have acquired *H. lauricola* through lateral transfer from *X. glabratus* and now function as secondary vectors. The US hosts and suscepts of laurel wilt disease include avocado and important species in forest ecosystems, including redbay, swampbay, and silkbay. Since laurel wilt spreads rapidly into new ecosystems and is difficult to control in avocado groves, more IPM tools are needed to manage the beetle vectors of *H. lauricola*. This study examined the efficacy of piperitone as a repellent for *X. glabratus* and native ambrosia beetle species in South Florida. Piperitone effectively repelled *X. glabratus* from traps baited with host-based lures but did not inhibit the attraction of native ambrosia beetles. The efficacy of piperitone was equal to that of verbenone; however, the lower cost of piperitone warrants its further investigation as a new repellent for potential incorporation into IPM programs for *X. glabratus*.

## Figures and Tables

**Figure 1 plants-12-02406-f001:**
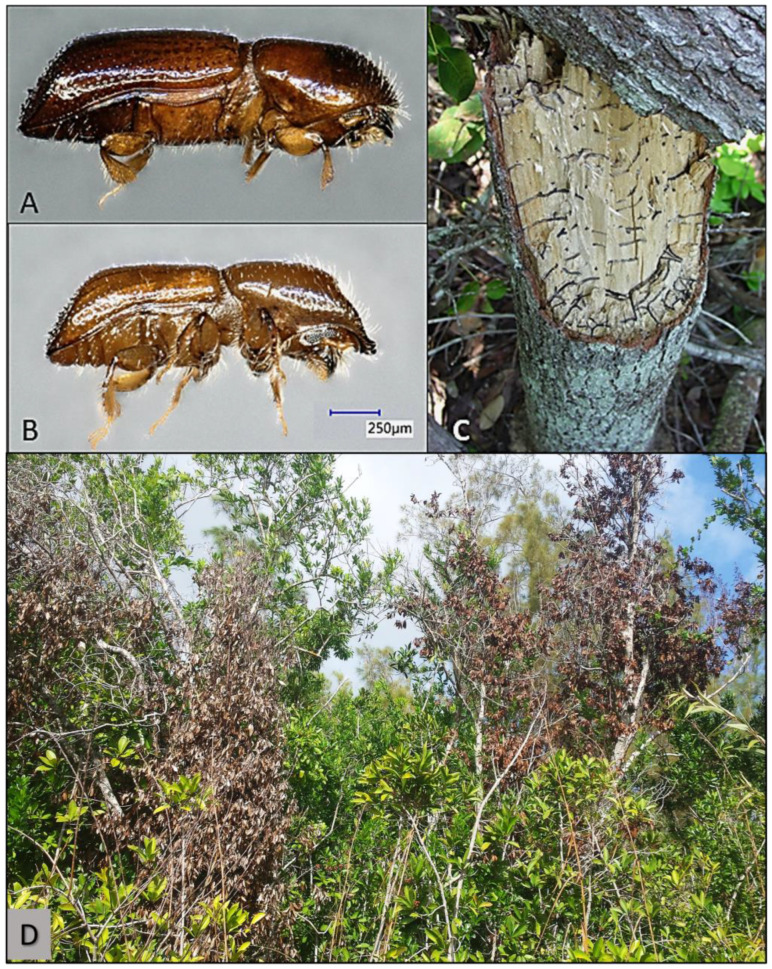
Redbay ambrosia beetle, *Xyleborus glabratus*, adult female (**A**), adult male (**B**), galleries in the trunk of a host swampbay tree, *Persea palustris* (**C**), and swampbay trees with laurel wilt in an Everglades tree island, Miami-Dade County, Florida, USA. (**D**). (Photo credits: (**A**,**B**)—T.I.N.; (**C**,**D**)—P.E.K.)

**Figure 2 plants-12-02406-f002:**
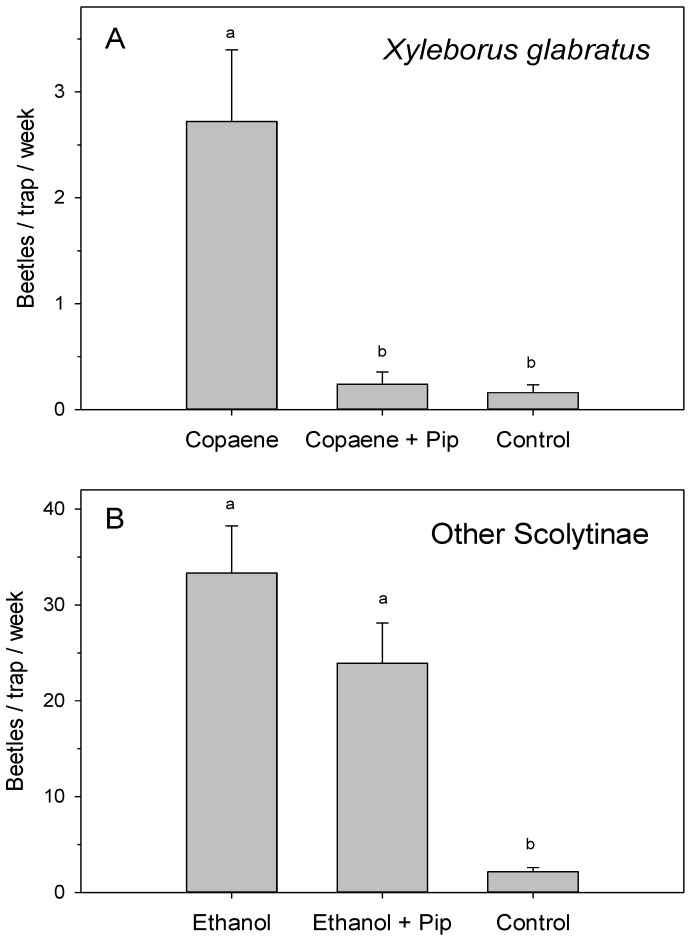
Mean (±SE) captures of female *Xyleborus glabratus* ((**A**), field test 1) and all other bark and ambrosia beetles ((**B**), field test 2) at a swampbay site with laurel wilt, Miami-Dade County, FL. The treatments for test 1 included an α-copaene lure, α-copaene lure plus piperitone, and a non-baited control trap. The treatments for test 2 included a low-release ethanol lure, ethanol lure plus piperitone, and a non-baited control trap. Both tests were conducted for 10 weeks. Bars topped with the same letter are not significantly different (Tukey HSD mean separation, *p* < 0.05).

**Figure 3 plants-12-02406-f003:**
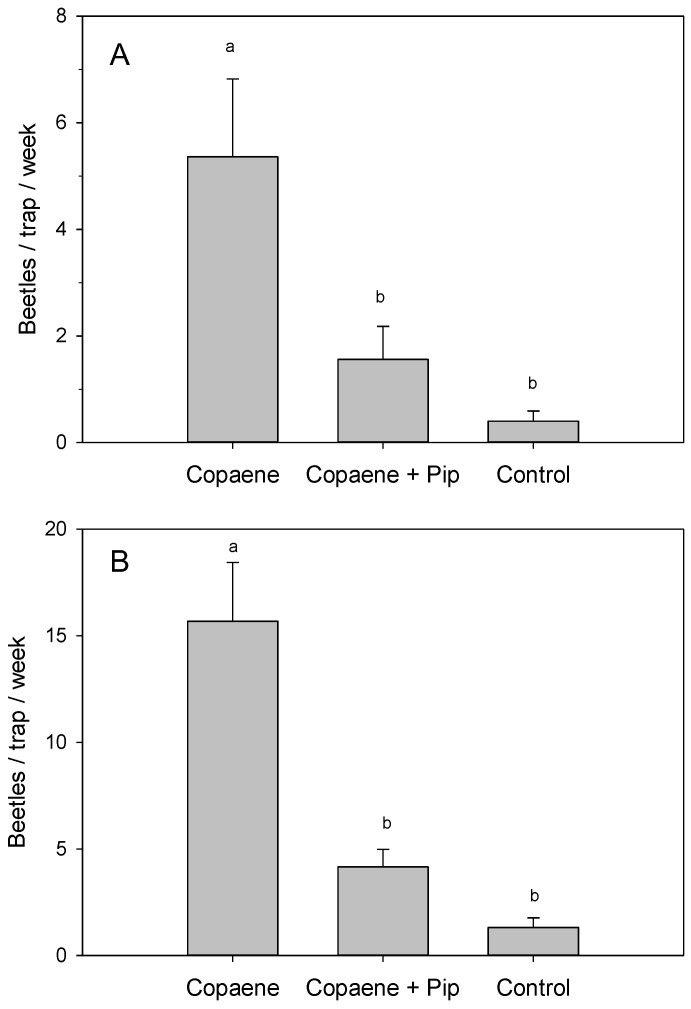
Mean (±SE) captures of female *Xyleborus glabratus* in field test 3 (**A**) and field test 4 (**B**), each conducted for 10 weeks at a swampbay site with laurel wilt, Broward County, FL. Treatments included an α-copaene lure, α-copaene lure plus piperitone, and a non-baited control trap. Bars topped with the same letter are not significantly different (Tukey HSD mean separation, *p* < 0.05).

**Figure 4 plants-12-02406-f004:**
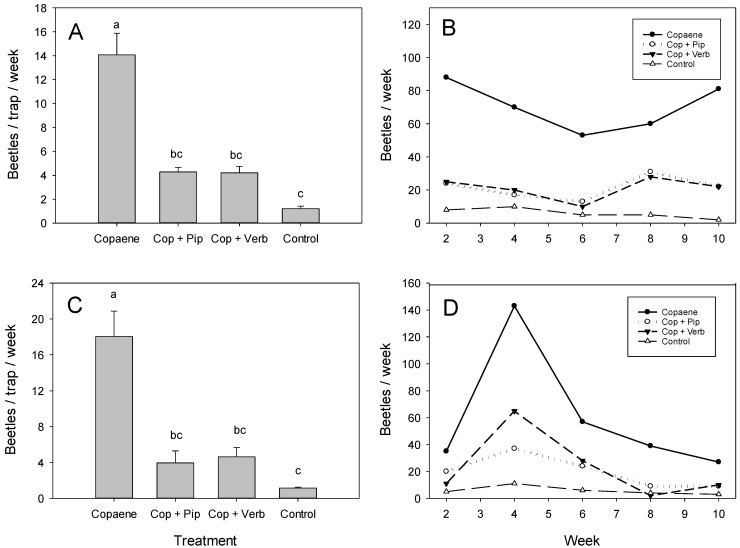
Captures of female *Xyleborus glabratus* in replicate 10-week field tests comparing the efficacy and longevity of piperitone and verbenone at a swampbay site with laurel wilt, Broward County, FL. Mean (±SE) captures (**A**) and summed weekly captures (**B**) in field test 5; mean (±SE) captures (**C**) and summed weekly captures (**D**) in field test 6. Treatments consisted of an α-copaene lure, α-copaene lure plus piperitone, α-copaene lure plus verbenone, and a non-baited control trap. For panels (**A**,**C**), bars topped with the same letter are not significantly different (Tukey HSD mean separation, *p* < 0.05).

**Table 1 plants-12-02406-t001:** Bark and ambrosia beetles captured in two 10-week field tests conducted at a swampbay site with laurel wilt, Miami-Dade County, Florida, USA.

Species			α-Copaene, Test 1	Ethanol, Test 2
Subfamily Scolytinae				
	Tribe Dryocoetini			
		*Coccotrypes carpophagus* (Hornung)	0	1
	Tribe Xyleborini			
		*Ambrosiodmus devexulus* (Wood)	7	12
		*Ambrosiodmus lecontei* Hopkins *	3	16
		*Premnobius cavipennis* Eichhoff	3	2
		*Theoborus ricini* (Eggers)	0	2
		*Xyleborinus andrewesi* (Blandford) *	8	50
		*Xyleborinus gracilis* (Eichhoff) *	14	22
		*Xyleborinus saxesenii* (Ratzeburg) *	33	448
		*Xyleborus affinis* Eichhoff *	17	116
		*Xyleborus bispinatus* Eichhoff *	2	22
		*Xyleborus glabratus* Eichhoff *	135	15
		*Xyleborus ferrugineus* (Fabricius) *	5	5
		*Xyleborus volvulus* (Fabricius) *	35	213
		*Xylosandrus compactus* (Eichhoff)	22	70
		*Xylosandrus crassiusculus* (Motschulsky) *	3	10
	Tribe Cryphalini			
		*Cryptocarenus heveae* (Hagedorn)	0	5
		*Hypothenemus* spp.	11	380
	Tribe Corthylini			
		*Corthylus papulans* Eichhoff	14	236
		Total	312	1625

* Species from which *Harringtonia lauricola*, the etiological agent of laurel wilt, has been isolated [55].

## Data Availability

The data are available from the authors upon request.

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
