# Peer review of "A New Repellent for Redbay Ambrosia Beetle (Coleoptera: Curculionidae: Scolytinae), Primary Vector of the Mycopathogen That Causes Laurel Wilt"

_plants, 2023, doi:10.3390/plants12132406_

Round 1

Reviewer 1 Report

Cloonan et al. describe a study to evaluate the effects of piperitone and verbenone on the attraction of the redbay ambrosia beetle and other ambrosia beetles. The study provides useful information and will be of interest a wide community. An important consideration is use of the term repellent. Since piperitone was tested alongside an attractant, the experimental design tested piperitone as interrupting the attraction of X. glabratus to an attractant. A repellent causes oriented movement away from a point source. The methods do not specifically demonstrate a repellent behavioral response, but instead demonstrate an interruption in the attraction to piperitone. Behavioral studies using a four arm olfactometer to demonstrate oriented movement away from piperitone would provide evidence for a repellent effect. Instead, the data demonstrate that piperitone interrupts the attraction to copaene. The manuscript should be revised to reflect this important distinction.

Overall, this is a great paper illustrating the need to help control X. glabratus and mitigate the spread of laurel wilt disease. The authors do a good job illustrating the potential for piperitone as an effective repellent. I’d suggest some minor restructuring of the paper to better convey ideas and help explain/reason for some of the experimental design. Methods, specifically experimental design and statistical analysis, could be better explained – for example, replications (ie, number of blocks) is not mentioned, no analysis is done by week leading the reader to assume weeks were pooled, but this is not discussed. Furthermore, the field tests (1-6) are not set up in a way to explain why they were done during these time frames and comparing or contrasting between some of these is confusing (one graph shows differences between FT1 and FT2, but different lures were used, whereas FT1, FT3-4 seem to be more similar). Some of the discussions also seem to be better placed in the introduction as a means to set up reasoning for experiments rather than explaining results. Finally, some results do not clearly convey through – the reduction of beetles using the repellent is described, but actual captures are not. With some minor revisions, this paper will be a good addition illustrating piperitone as a part of a toolkit in managing ambrosia beetles.

L13:  “…. in the Lauraceae family”

L22: α-copaene-baited traps comes out of the blue – tie this in better

L36-37: citation for “X. glabratus can attack healthy, unstressed trees”

Ln 67-78: although this paragraph is informative, it pulls away from the focus on X. glabratus. Can you add a sentence or two explaining why X. glabratus is still the primary concern rather than these other species (ie, X. bispinatus?) perhaps the population numbers are less of a concern or their seasonality, etc?

L87-93: reword to better convey your message. Why is Mexico of concern? Is it moving that direction from Georgia?

L94-103: Reword to relate verbenone and bark beetles. Maybe add a citation or a bit more info on expenses of verbenone and piperitone in order to justify this as a cheaper alternative? Is it cents cheaper, dollars or tens of dollars?

L104-115: same as above paragraph, reword or restructure to better explain piperitone’s potential as a repellant for X glabratus. Why was it tested on these other beetle species? How does it work (ie, anti-aggregation?). Why is it cheaper (ie, is it synthetic or easier to isolate?). I’d also recommend a better transition or separating into two paragraphs from talking about piperitone to your summary statement – as it is, it sounds like another study with piperiton that you plan to discuss until the near end of that section

L122-134: repellant longevity? There is no mention of time, but these experiments were set up over 10 weeks. Assuming weeks were pooled, this isn’t mentioned in methods and repeated measures are also not mentioned. While the number of reduced captures is informative, including the control or lure only trap captures is needed in order to understand the reduced captures.

L135: No mention yet of a control trap in methods. Also, Field test 1 and 2 were set over consecutive dates, not synchronous. This should be highlighted in the figure caption

Table 1: Here, you organize the data into lure type rather than field test 1 or 2, which makes the information you’re showing a bit confusing. Perhaps clarify the respective lure with sampling dates (I believe this is important as the ethanol test was conducted during a warmer 10 week time frame than the Copaene) and field test in captions and also clarify if these are captures in all traps (lure, lure + repellant and control).

L158: see comments on L122-134, While the number of reduced captures is informative, including the control or lure only trap captures is needed in order to understand the reduced captures.

Figures 2-3: another important aspect is that X. glabratus numbers increase overall in Field test 1 to FT2 and then FT3 – this seems to indicate time of trapping/experiments may also affect trap captures – ie, warmer time frames have larger numbers of beetles.

Figure 4: Here is first mention of capturing only female X. glabratus – are males non-fliers? Into identifies females as carriers/inoculators of symbionts, but is anything known on males? Are they not of concern?

L216: “some protection” – introduction made a point of prevention as means of mitigating an already infected tree wasn’t reliable?

L219: first mention of endemic laurel wilt – can you explain the bridge between endemic laurel wilt and the introduction of X. glabratus in 2002?

L218-233: This paragraph starts our with results in 3 of the 6 field tests- what about the other field tests?, This then goes into SPLAT and verbenone efficacy in repelling X. glabratus, maybe make the connection a little more clear, ie, perhaps the reduced captures with piperitone could also lead to higher tree survival rates?

L234-236: why are field trials 5 and 6 separated from 1, 3 and 4 when discussing reduced captures?

L247: Most ambrosia …. Including those not repelled by piperitone in this study…. I don’t believe this is mentioned at all prior, that some species were not repelled?

L257-289: These almost seems more appropriate in introduction rather than discussion. This does a better job setting up the experiment than it does explaining the results

L317-326: Height of traps?

L343: set of traps? How many?

L346: should tests be “traps”?

L347: trap positions were rotated sequentially… what does this mean? How does this minimized positional effects on beetle captures? (also L367)

General comment from methods – I think its great you’re describing the stage of laurel wilt observed when deploying the traps, but if it can take as little as 4-6 weeks to kill these trees, how might this effect the beetles and/or their flight – are they more likely to seek out the lures during a “peak flight” or are you coming in a bit late once laurel wilt has been observed at all? Discussing this in the introduction with the disease might be good in order to make the methods a bit more clear.

Also, can you clarify in methods about what a “block” consists of, replications and number of traps? Were these averaged for each “cluster of symptomatic swambay trees”? These clusters sound like your unit of measurement and that you may have pseudo-replications. It’s difficult to understand how this circle was a RCBD if you formed a circle, then a rectangle and why these were rotated.

L379-381: How is a t-test comparing longevity of repellency? Analysis was done within each week, not together over the 10 weeks?

Reviewer 2 Report

Congratulations on the manuscript!

Please correct "Journal of Pest Sciience" in row 603.

specific comments: 1. What is the main question addressed by the research? Xyleborus glabratus is a vector of laurel wilt, a lethal disease of trees in the Lauraceae affecting both natural habitats and plantations. Repellent volatiles could broaden our understanding of the chemical communication of X. glabratus and could help to improve the present plant protection products. 2. Do you consider the topic original or relevant in the field? Does it address a specific gap in the field? Piperitone was demonstrated to repel to deter attacks by X. glabratus and other ambrosia beetles. This achievement is a novelty that helps understand the orientation of the X. glabratus. It might be a cheaper active or a combination partner of better repellent formulations for pest management practice. 3. What does it add to the subject area compared with other published material? It is a novel repellent for this insect species and other related species. 4. What specific improvements should the authors consider regarding the methodology? What further controls should be considered? The manuscript is acceptable for publication. 5. Are the conclusions consistent with the evidence and arguments presented and do they address the main question posed? Yes, they do. 6. Are the references appropriate? The references are appropriate. There is a large number of publications altogether 106 of them are cited. 7. Please include any additional comments on the tables and figures. The tables and figures are fine.

Reviewer 3 Report

Reviewer: 1

Ms. Ref. No.: Plants-2384231

Authors: Kevin R. Cloonan *, Wayne S. Montgomery, Teresa I. Narvaez, And Paul E. Kendra

Specific notes:

TITLE

Nothing to comment

ABSTRACT

Nothing to comment

INTRODUCTION

All paragraphs in the introduction section must be justified

Line 34: Xyleborus glabratus should appear abbreviated

Line 48: Harringtonia lauricola should appear abbreviated

Lines 69-71: Do not repeat the genera of the species.

Line 97: Pinus contorta var. latifolia, “var.” should not appear in italics.

Line 115: The objectives of the study do not appear or are not clearly described

RESULTS

All paragraphs in the results section must be justified

Lines 126-127: in the scientific results if there are no significant differences in the results, they are not described in the text. Eliminate in the manuscript in all those text of results in that there are no significant differences.

Line 145: Harringtonia lauricola should appear abbreviated

Line 146: Xyleborus glabratus should appear abbreviated

DISCUSSION

All paragraphs in the discussion section must be justified

Line 211: ”40km/year in 2016 to 24km/yr in 2021”, please, homogeneity in the units

Line 281: Do not repeat the genera of the species.

MATERIAL AND METHODS

All paragraphs in the matherial and methosd section must be justified

CONCLUSIONS

The paragraphs in the conclusion section must be justified

Nothing to comment

FIGURES

Nothing to comment

TABLES

Nothing to comment

REFERENCES

Nothing to comment

Round 2

Reviewer 3 Report

The authors have made the suggested changes, Thank you very much.

Author Response

Thank you very much.